# Grasp the Key Takeaways from Source Domain for Few Shot Graph Domain Adaptation

## Abstract

Graph Neural Networks (GNNs) have achieved remarkable success in node classification tasks on individual graphs. However, existing GNNs trained within a specific domain (*a.k.a.*, source domain) frequently exhibit unsatisfied performance when transferred to another domain (*a.k.a.*, target domain), due to the domain gap. To tackle this issue, Few Shot Graph Domain Adaptation (FSGDA) is introduced to the node classification task, facilitating knowledge transfer from a fully labeled source graph to a target graph with minimal annotations for each class. An intuitive solution is directly training the GNN with labeled source and target samples together. Nevertheless, there are two issues in this procedure: (1) When the annotations on the target domain used for training are extremely sparse, the GNN performance may significantly be damaged by nodes with the source-domain bias not aligning with the target-domain distribution. (2) Apart from the biased nodes, the low-value nodes among the remaining nodes impede the GNN learning for the core nodes, like the limited target training nodes. To address the above issues, we propose a new method for FSGDA, named GraphInflu, whose core idea is to grasp the key takeaways from the source domain to facilitate the adaptation process. It contains two characteristic modules, including the Supportive Node Selector and the Soft Logic-Inspired Node Reweighting. The former aims to identify the most influential set of source nodes based on their contribution to improving performance on target nodes. The latter further focuses more on the core nodes in the selected influential set, which closely align with the target nodes especially those presenting challenging predictions. Extensive experiments validate the efficacy of GraphInflu by overcoming the current state-of-the-art methods. Our code is available at https://anonymous.4open.science/r/GraphInflu-E8E7.

## CCS Concepts

• **Computing methodologies → Machine learning**.

## Keywords

Graph Domain Adaptation, Few-shot Learning

## 1 Introduction

Node classification using Graph Neural Networks (GNNs) is a fundamental yet challenging task in a multitude of applications, such as citation networks [14], social networks [7], and webpage networks [15]. While GNNs have demonstrated significant success in domain-specific tasks (*a.k.a.*, **source domain**), their performance often degrades when applied to a different domain (*a.k.a.*, **target domain**) due to the **domain gap**—a divergence in data distributions across domains [22, 23, 33, 34]. This gap is particularly pronounced in scenarios like cross-domain citation networks, where structural and feature differences between networks limit the direct transferability of GNN models [4, 14, 30].

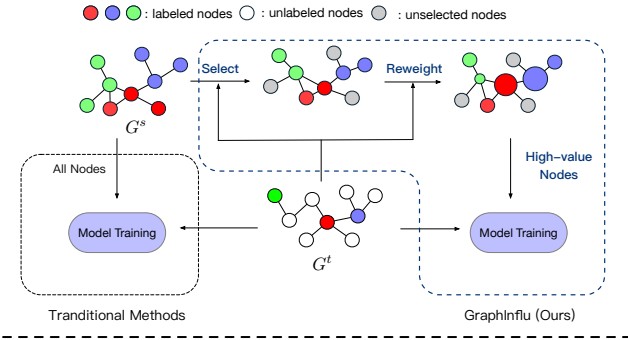

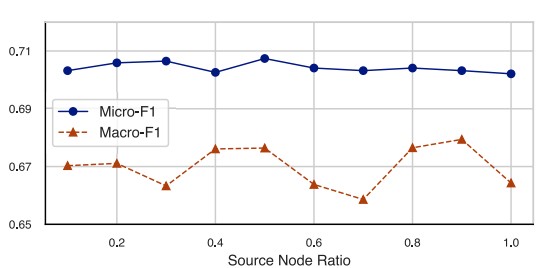

**Figure 1:** *Top*: **Differences between our method and traditional methods to address Few-shot Graph Domain Adaptation (FSGDA). Our key point is not all source-domain nodes facilitate model training on FSGDA and we may just need to utilize the necessary ones.** *Bottom*: **Performance of GCN in the cross-graph node classification scenario: ACMv9 ⇒ DBLPv7.**

A straightforward solution for **Graph Domain Adaptation (GDA)** would be to manually annotate a large number of target domain nodes and fine-tune the GNN. However, this approach is impractical due to the time and effort required for large-scale node annotation. To overcome this bottleneck, **Few-Shot Learning (FSL)** has emerged as a promising alternative, aiming to transfer knowledge from the source domain to the target domain using only a few labeled nodes in each class [32, 40]. Prior methods [5, 35] typically train GNNs on both the source and target domains, leveraging available annotations. However, the extreme sparsity of labeled nodes in the target domain introduces two major challenges: **(1) Adverse Node Interference.** If we keep the number of the target-domain training samples and increment the training rate of the source network from zero, we find that the GNN performance on the target domain rapidly saturates and then subsequently stabilizes. Taking ACMv9 [27] (source network) and DBLPv7 [27] (target network) as an example, the performance of the Graph Convolutional Network (GCN) [14] trained on them are shown in Figure 1. This rapid stabilization may stem from the mutual counterbalance between "beneficial" and "harmful" samples within the

training dataset. Specifically, part of source-domain samples (*adverse node*) carry biases that deviate from the target domain distribution, such as differences in feature distributions and neighborhood structures. As training progresses, the GNN adeptly acquires these source-domain biases by the neighbor-node aggregation, leading to significant damage in model accuracy. Such negative impact is balanced by the positive impact brought by the *supportive nodes*, consisting of the remaining source-domain samples and the extremely limited target-domain labeled samples. This balance results in the stagnation of model performance. Given the constraint of not being able to access more target-domain labeled samples, the key challenge to disrupt this balance and further enhance the model lies in how to filter out adverse nodes. **(2) Core Nodes Inundated.** Even effectively removing adverse nodes, the remaining subset may only contain a small number of high-value core samples, including extremely limited annotated target-domain nodes, a few source-domain nodes closely aligned with the target domain distribution, and high-quality nodes containing richer classification knowledge. However, these limited core nodes are susceptible to being overwhelmed by more low-value supportive nodes, hindering the GNN's thorough learning of underlying patterns. Given the limitation in extending annotations for target-domain nodes, identifying and elevating the importance of core nodes in the source domain becomes a pertinent issue worthy of consideration.

To tackle the above issues, we propose the **GraphInflu** model for few-shot graph domain adaptation by grasping the key takeaways from the source domain, as shown in Figure 2. Our method is specifically designed with two phases to address the corresponding challenges. **Phase 1**: We aim to establish a supportive node selector that scores the source nodes according to their contribution to reducing loss on labeled target nodes. To efficiently compute the contribution score, we approximate loss reduction by comprehensively analyzing gradients of the GNN model on the source nodes and the target nodes. In this way, we develop a contribution score function to identify supportive source nodes. Given the score matrix obtained from the score function, we further incorporate a class-balanced sampling strategy to avoid selection bias toward certain classes. **Phase 2**: We delve into assessing the significance of each selected supportive source domain data. Specifically, on the one hand, we introduce perturbations through adversarial learning to gauge the stability of GNN model predictions for target nodes; on the other hand, we calculate entropy to evaluate the certainty of GNN predictions for target nodes. By computing the distances between node-centric subgraphs, we evaluate the proximity of source-domain nodes to target-domain nodes that exhibit unstable or uncertain predictions. We posit that source-domain nodes resembling these challenging target-domain nodes possess greater learning value.

Our contributions are summarized as follows:

- We study the Few Shot Graph Domain Adaptation (FSGDA) and uncover the substantial negative impact of adverse source nodes when the annotated target nodes are scarce. To the best of our knowledge, we delve into early exploration from the view of the key source-node retrieval for the FSGDA.

- We propose a novel method named **GraphInflu**. We filter out adverse source nodes based on backward gradients and subsequently enhance the importance of the high-value nodes via the graph similarity calculation and the stability estimation based on a FOL framework.

- We conduct extensive experiments to demonstrate the effectiveness of our proposed method, and our framework achieves a new state-of-the-art result.

## 2 Related Work

Graph domain adaptation (GDA) extends traditional domain adaptation (DA) to graph-structured data [19, 21, 24, 34, 42], where features and labels are interconnected due to the graph structure. Recent studies aim to combine graph models with domain adaptation techniques to learn domain-invariant representations. These methods often use adversarial learning or minimize the distance between representations in the source and target domains. ACDNE [22] applies adversarial domain adaptation to make node representations invariant across networks. UDAGCN [33] further introduces an inter-graph attention mechanism combined with adversarial training. MFFReg [39], a more recent approach, uses graph spectral regularization to improve the transferability of Graph Neural Networks (GNNs).

In addition to the existing unsupervised graph domain adaptation (UGDA) methods, AdaGCN [5] and SemiGCL [35] address scenarios with limited labeled nodes in the target graph, closely aligning with our research focus. AdaGCN minimizes domain discrepancy using the Wasserstein distance, while SemiGCL combines graph contrastive learning with minimax entropy training to generate discriminative node representations. However, to the best of our knowledge, these methods primarily train the model using source nodes and a few labeled target nodes with a standard cross-entropy loss. These approaches treat all source nodes equally, overlooking the unique characteristics of individual source nodes. Another related approach in graph domain adaptation is test-time graph adaptation [3, 31], which adjusts graph data during the testing phase. In contrast, our method focuses on the training phase, where we identify influential source nodes and reweight them.

Recently, few-shot learning on graphs has been proposed to tackle the issue of limited labeled data in real-world scenarios [26, 41]. Existing studies on few-shot node classification can be divided into two categories: (1) metric-based methods [25, 38], which primarily classify new nodes by calculating the Euclidean distance between node embeddings and class prototypes; (2) optimization-based methods [8, 43], which aim to learn a better initialization of model parameters that can be updated by a few gradient steps for new tasks.

## 3 Preliminaries

### 3.1 Notations

Let $G = (V, E, A, X, Y)$ represent a graph, where $V$ is the node set, $E$ is the edge set, and $A$ is the adjacency matrix. The number of nodes and edges are denoted by $n$ and $m$, respectively. $X$ is the node feature matrix, and $Y$ represents the node labels.

In the Few Shot Graph Domain Adaptation (FSGDA) scenario, we have a fully labeled source graph $G^s = (V^s, E^s, A^s, X^s, Y^s)$

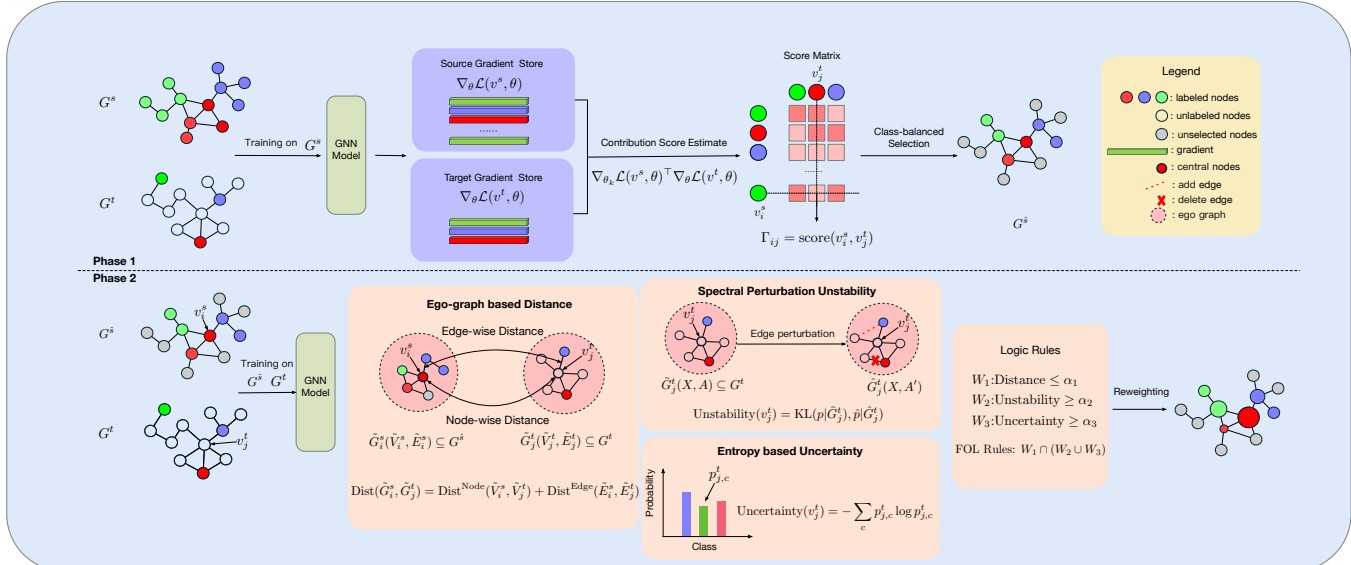

**Figure 2: The architecture of GraphInflu consists of two phases: (a) *Supportive Node Selector via Gradients* identifies the most influential source nodes through gradient matching between gradient information of source nodes and few-shot labeled target nodes; and (b) *Soft Logic-Inspired Node Reweighting* further assigns greater importance to selected source nodes that closely align with the target domain, particularly where challenging target nodes exist. It estimates the *ego-graph based distance* to ensure that reweighted source nodes are sufficiently close to certain target nodes. The *unstability* of target nodes is evaluated using the KL divergence between the original ego graphs and their corresponding adversarially perturbed versions. The *uncertainty* of target nodes is estimated through entropy calculation. Subsequently, a First-Order Logic (FOL) framework is incorporated to balance these metrics, facilitating the reweighting of source nodes.**

with $n^s$ nodes, and a partially labeled target graph $G^t = (V^t, E^t, A^t, X^t, Y^t)$ with $n^t$ nodes. The source graph $G^s$ contains a fully labeled node set $V^{s,l}$, where $n^{s,l} = n^s$. In the target graph $G^t$, the nodes are divided into labeled nodes $V^{t,l}$ and unlabeled nodes $V^{t,u}$, such that $n^{t,l} + n^{t,u} = n^t$, with $n^{t,u}$ is significantly larger than $n^{t,l}$. Typically, there are only a few labeled nodes per class in the target graph, for instance, five labels per class (5-shot). We assume the label space is shared between the source and target domains, with $C$ denoting the number of classes. The objective of FSGDA is to train a model $g$ that performs well on the target graph, leveraging the labeled source nodes and the limited labeled target nodes.

### 3.2 General Objective Function of FSGDA

The general objective function of FSGDA can be formulated as follows:

$$\mathcal{L}_{\text{FSGDA}} = \mathcal{L}_{\text{cls}} + \mathcal{L}_{\text{align}} \left[+\mathcal{L}_{\text{other}}\right] \qquad (1)$$

where $\mathcal{L}_{\text{cls}}$ denotes the cross-entropy loss function for the node classification task using labeled nodes. This term can be further divided into two parts based on the origin of nodes: $\mathcal{L}_{\text{cls}} = \mathcal{L}_{\text{cls}}^s + \mathcal{L}_{\text{cls}}^t$. And the $\mathcal{L}_{\text{align}}$ term represents the domain alignment loss, which can be implemented using techniques such as maximum mean discrepancy (MMD) [11, 37] or adversarial training mechanisms [1, 9]. The last term, $\mathcal{L}_{\text{other}}$, is optional and usually denotes self-supervised methods, such as graph contrastive learning techniques employed in both source and target graphs [28, 35].

## 4 Method

In this section, we introduce a novel method GraphInflu, which aims to identify the most influential source nodes that better align with the target domain and further enhance the importance of these high-value nodes. To address this, as shown in Figure 2, our method is divided into two phases: (1) **supportive node selector via gradients**, and (2) **soft logic-inspired node reweighting**.

**In phase 1**, we train a GNN model on the source graph and compute the loss for all labeled nodes during the training process. By performing backpropagation, we obtain the gradient stores for both the source and target nodes. Furthermore, we deduce that the change in loss for labeled target nodes can be measured by the inner product of the gradient features from the two networks. Based on this, we construct a **contribution score function** for the source nodes using these two sets of gradient features. Finally, we apply a class-balanced sampling strategy based on the contribution score to select supportive data. **In phase 2**, we train a new GNN model on both the source and target graphs. During training, we place greater emphasis on source nodes whose distribution closely aligns with the target nodes, particularly the challenging target ones. The representativeness metric is evaluated based on the distance between ego graphs centered specific nodes from both domains, consisting of both node-wise and edge-wise distances. The difficulty metric consists of two components: unstability and uncertainty of specific target nodes. For unstability, we construct an adversarially perturbed ego graph by maximizing the

spectral distance. The unstability metric is computed using the KL divergence between the original prediction (based on the original ego graph) and the perturbed prediction (based on the perturbed graph). For uncertainty, we estimate the model's uncertainty via the entropy of its predictions. These three metrics are then integrated into a First-Order Logic (FOL) framework to compute an importance weight for each source node.

## 4.1 Supportive Node Selector via Gradients

As mentioned in introduction, a certain proportion of source domain nodes (referred to as adverse data) introduce biases that deviate from the target domain distribution. As the model trains, these biases are enhanced and thus the degrades model performance on the target domain. With limited labeled data in the target domain, here we aim to identify the influential source nodes and filter out adverse ones to improve model performance.

Here we emphasize the effective utilization of the few labeled target nodes. We offer a unique perspective by viewing source graph nodes as training data and correspondingly labeled target nodes as validation data. This allows the performance on these "validation" data (labeled target nodes) to provide insights into model's generalization ability on the target distribution. Consequently, the problem can be transformed into selecting the most influential source nodes to achieve better performance on these limited labeled target ones.

**Contribution score function.** We consider using gradient features from both labeled source nodes and target nodes to construct our contribution score function. Consider a node classification model $g = h \circ f$ trained on the loss $\mathcal{L}$. Here $f$ denotes the feature extractor and $h$ is the node classification head. The overall parameters are represented by $\theta$. Notably, the training process relies exclusively on the source graph, while the few labeled target nodes are solely used for evaluation. In this way, we can estimate first-order Taylor expansion of loss on the labeled target node $v_j^t$ at the $(k + 1)$-th update of the model.:

$$\mathcal{L}(v_j^t, \theta_{k+1}) \approx \mathcal{L}(v_j^t, \theta_k) - \eta_k \nabla_{\theta_k} \mathcal{L}(v_j^t, \theta_k)^\top \nabla_{\theta_k} \mathcal{L}(v_i^s, \theta_k) \quad (2)$$

where $v_i^s \in V^{s,l}$ and $v_j^t \in V^{t,l}$ represent the source node and target node, respectively. $\eta_k$ denotes the learning rate at step $k$. $\nabla_{\theta_k}(\cdot)$ signifies the gradient of the loss function with respect to the model weights $\theta_k$. Furthermore, we can formulate the following loss reduction extent at step $k$:

$$\mathcal{L}(v_j^t, \theta_k) - \mathcal{L}(v_j^t, \theta_{k+1}) \approx \eta_t \nabla_{\theta_k} \mathcal{L}(v_j^t, \theta_k)^\top \nabla_{\theta_k} \mathcal{L}(v_i^s, \theta_k) \quad (3)$$

We can observe that the inner product of gradient features $\nabla_{\theta_k} \mathcal{L}(v_j^t, \theta_k)^\top \nabla_{\theta_k} \mathcal{L}(v_i^s, \theta_k)$ measures the change in the loss on labeled target nodes. The magnitude of the change in loss reflects the extent to which the source node contributes to the performance on the corresponding target node. Therefore, we can utilize this item to define the following contribution score function $\text{score}(\cdot, \cdot)$ through aggregating multiple learning steps:

$$\text{score}(v_i^s, v_j^t) = \sum_k \eta_k \nabla_{\theta_k} \mathcal{L}(v_j^t, \theta_k)^\top \nabla_{\theta_k} \mathcal{L}(v_i^s, \theta_k) \quad (4)$$

The above formulation illustrates the contribution score of a source node in relation to a specific target node. Furthermore, we can derive the contribution score matrix $\Gamma \in \mathbb{R}^{n^{s,l} \times n^{t,l}}$, where $\Gamma_{ij} = \text{score}(v_i^s, v_j^t)$.

**Class-balanced selection strategy.** In practice, relying solely on the highest score for selection from Eq. (4) tends to construct a class biased source node set. One possible reason is that the semantic representations of certain classes have a smaller domain gap, leading to higher scores for source nodes from those classes.

To address this issue, we employ a class-balanced selection strategy. For a given source node $v_i^s$, we aggregate the contribution score based on the label of few-shot target nodes. In this way, we can obtain a class-wise contribution score matrix $\hat{\Gamma} \in \mathbb{R}^{n^{s,l} \times C}$:

$$\hat{\Gamma}_{ij} = \frac{\sum_{v_m^t \in V^{t,l}} \mathbb{I}(y_m^t = c_j) \cdot \Gamma_{ij}}{\sum_{v_m^t \in V^{t,l}} \mathbb{I}(y_m^t = c_j)} \quad (5)$$

where $\mathbb{I}$ is an indicator function that returns 1 when the condition is satisfied and 0 otherwise. Here $\hat{\Gamma}_{ij}$ denotes the contribution score of source node $v_i^s$ belonging to class $c_j$. Assume we select a total of $Q$ source nodes from $V^s$. For a specific class $c_i$, we choose $\frac{(1+\beta)Q}{C}$ samples with the highest scores under class $c_i$, where $\beta > 0$ facilitates the selection of additional candidates. This adjustment addresses the potential overlap among class-wise candidates; for example, a single node may exhibit high contribution score in both class $c_i$ and class $c_j$. Finally, we can obtain a class-balanced supportive source node set $V^{\hat{s}}$ via employing $\Gamma$ and our sampling strategy.

## 4.2 Soft Logic-Inspired Node Reweighting

Even with supportive nodes identified by the above score function, the core source nodes can easily be overshadowed by low-value nodes, limiting the GNN's ability to fully capture the underlying patterns. Given the limited availability of target domain labels, we further emphasize the importance and uniqueness of core source nodes closely aligned with the target nodes, especially the challenging target ones.

The previous section primarily emphasizes the use of labeled target nodes to identify the supportive source nodes. However, the abundant unlabeled target nodes also provide valuable information for domain adaptation. In this section, we explore the role of source nodes in the domain adaptation process by effectively utilizing these numerous unlabeled target nodes. Here, we consider two kinds of challenging nodes: unstable and uncertain target nodes, as they contribute more to domain adaptation when properly addressed.

Given a source node $v_i^s$, we first find its $K$ nearest target nodes $\mathcal{A}_i = \{v_j^t\}_{j=1}^K$ in the representation space. The unstability and uncertainty of these $K$ target nodes are then estimated to dynamically adjust the importance of the corresponding source node $v_i^s$.

**Ego-graph based distance.** Given the structural nature of graphs, we assess the similarity between subgraphs centered on nodes from two domains, which encompasses both node-wise and edge-wise distances. Specifically, for a source node $v_i^s$ and a target node $v_j^t$, we extract their respective ego graphs, denoted as $\tilde{G}_i^s(\tilde{V}_i^s, \tilde{E}_i^s)$ and $\tilde{G}_j^t(\tilde{V}_j^t, \tilde{E}_j^t)$. An ego graph represents the induced subgraph surrounding a specific node. We use $\tilde{V}_i^s$ and $\tilde{V}_j^t$ to denote the induced sets of nodes, and $\tilde{E}_i^s$ and $\tilde{E}_j^t$ to represent the induced sets of edges.

The number of nodes in $\tilde{G}_i^s$ and $\tilde{G}_j^t$ is represented as $n_1$ and $n_2$, respectively,

The node-wise distance can be calculated using metrics based on two node sets, $\tilde{V}_i^s$ and $\tilde{V}_j^t$. In this context, we employ the Wasserstein distance [20] for node-wise distance $D^{\text{Node}}(\tilde{V}_i^s, \tilde{V}_j^t)$, as follows:

$$D^{\text{Node}}(\tilde{V}_i^s, \tilde{V}_j^t) = \min_{\gamma \in \Pi(\mu_i, \mu_j)} \sum_{a \in \tilde{V}_i^s} \sum_{b \in \tilde{V}_j^t} \gamma_{ab} \cdot c(x_a, x_b) \qquad (6)$$

where $\Pi(\mu_i, \mu_j) = \left\{ \gamma \in \mathbb{R}_+^{n_1 \times n_2} \mid \gamma \mathbf{1}_{n_1} = \mu_i, \mathbf{1}_{n_2}^T \gamma = \mu_j \right\}$ denotes the valid transport plan $\gamma$. Here, $\mathbf{1}_{n_1}$ and $\mathbf{1}_{n_2}$ are column vectors of ones with $n_1$ and $n_2$ entries, respectively. Here for simiplicity, we use $a$ and $b$ to represent the nodes $v_a^s$ and $v_b^t$, respectively. $x_a$ and $x_b$ denotes the node embedding. And $c(x_a, x_b)$ denote the cost to move a distribution to another, here we choose the commonly used squared Euclidean distance, defined as $c(x_a, x_b) = \|x_a - x_b\|^2$.

Besides the node-wise distance between two ego graphs, we further incorporate the edge-wise distance to capture structure similarity. Following the same notions in Eq. (6), we employ Gromov-Wasserstein [2] to calculate distances $D^{\text{Edge}}(\tilde{E}_i^s, \tilde{E}_j^t)$ between the edge sets $\tilde{E}_i^s$ and $\tilde{E}_j^t$:

$$D^{\text{Edge}}(\tilde{E}_i^s, \tilde{E}_j^t) = \min_{\hat{\gamma} \in \Pi(\mu_i, \mu_j)} \sum_{(a,c) \in \tilde{E}_i^s} \sum_{(b,d) \in \tilde{E}_j^t} \hat{\gamma}_{ab} \cdot \hat{c}(x_a, x_c, x_b, x_d) \qquad (7)$$

where $\hat{c}(x_a, x_c, x_b, x_d) = \|\cos(x_a, x_c) - \cos(x_b, x_d)\|$ is the cost function between edges from different graphs. Here, cos refers to cosine similarity, defined as $\cos(x, y) = \frac{x \cdot y}{\|x\|\|y\|}$. The final ego-graph based distance $d_{ij}$ are formulated as follows:

$$d_{ij} = D^{\text{Node}}(\tilde{V}_i^s, \tilde{V}_j^t) + D^{\text{Edge}}(\tilde{E}_i^s, \tilde{E}_j^t) \qquad (8)$$

**Unstability estimate**. The unstability of the model's predictions on samples can be assessed by measuring the inconsistency between predictions on the samples and their corresponding virtual adversarial samples [10]. In this part, we first construct an adversarial sample by maximizing the spectral distance between the original graph and the perturbed graph, centered around specific nodes. And then we utilize the Kullback-Leibler (KL) divergence between the predictions from the original graph and the perturbed graph to estimate unstability.

For an ego graph $\tilde{G}_j^t$ centered at node $v_j^t$, we use $\mathcal{G}(X, A)$ with feature matrix $X$ and adjacency matrix $A$ to represent $G_j^t$ to simplify symbols. The normalized Laplacian matrix $L$ is defined as $L = I_n - D^{-\frac{1}{2}} A D^{-\frac{1}{2}}$, where $D$ is the diagonal degree matrix with entries $D_{ii} = \sum_{j=1}^{m} A_{ij}$. Furthermore, the edge decomposition of the Laplacian matrix $L$ can be expressed as $L = U \Lambda U^\top$, where $\Lambda = \text{diag}(\lambda_1, \ldots, \lambda_n)$ is the diagonal matrix of eigenvalues, and $U$ is a unitary matrix.

We generate edge perturbations $\Delta \in [0,1]^{n \times n}$ by maximizing the spectral distance between the original graph $\mathcal{G}(X, A)$ and perturbed graph $\mathcal{G}'(X, A')$. The perturbed adjacency matrix $A'$ is formulated as follows:

$$A' = \Delta \circ (\mathbf{1} - A) + (\mathbf{1} - \Delta) \circ A \qquad (9)$$

Here, $\Delta_{ij} = 1$ indicates a flip operation: if $A_{ij} = 1$, the edge is deleted; if $A_{ij} = 0$, the edge is added. The symbol $\circ$ means element-wise multiplication. The optimal perturbed matrix $A'$ can be optimized through the spectral distance $D_{\text{spec}}$, which is defined as $D_{\text{spec}} = \|g_\phi^*(\Lambda) - g_\phi^*(\Lambda')\|$, where $g_\phi^*$ represents the graph filter parameterized by $\phi$. Following the approach in [18], $g_\phi^*$ can be approximated using the first-order approximation of the Chebyshev polynomials [13], yielding $g_\phi^* \approx \|I_n - \Lambda\|$. Thus, $D_{\text{spec}}$ can be approximated by the formulation:

$$D_{\text{spec}} \approx \|(I_n - \Lambda) - (I_n - \Lambda')\| = \|\Lambda - \Lambda'\| \qquad (10)$$

Consequently, we can conclude our objective is to achieve an optimal edge perturbation $\Delta$ by solving the following optimization problem:

$$\Delta_0 = \arg\max_\Delta \|\Lambda - \Lambda'\| \qquad (11)$$

This formulation can be effectively solved using gradient descent. Once $\Delta$ is obtained, we can define the unstability of the target node $v_j^t$ using the Kullback-Leibler divergence between the original ego graph $\mathcal{G}$ and the corresponding adversarially perturbed ego graph $\mathcal{G}'(X, A')$ as follows:

$$\hat{w}_{j,\text{uns}}^t = \text{KL}(P(p_j^t | \mathcal{G}(X, A)), P(\tilde{p}_j^t | \mathcal{G}'(X, A'))) \qquad (12)$$

where $p_j^t$ and $\hat{p}_j^t$ represent the predictions under $\mathcal{G}$ and $\mathcal{G}'$, respectively.

**Uncertainty estimate**. The uncertain target nodes have low prediction confidence yet may be informative to the target domain. Here we employ the entropy to calculate the uncertainty of samples. For a given target node $v_j^t$, we extract its representation $z_j^t$ from the last layer of the feature encoder $f$, and obtain its prediction $p_j^t$ after passing through the classifier head $h$. The entropy for unlabeled target node $v_j^t \in V_{u,l}^t$ can be formulated as follows:

$$\hat{w}_{j,\text{unc}}^t = \text{Entropy}(v_j^t, \theta) = -\sum_c p_{j,c}^t \log p_{j,c}^t \qquad (13)$$

where $p_{j,c}^t$ denotes the prediction probability of $v_j^t$ belong to the class $c$.

**Node reweighting with FOL**. As illustrated above, we aim to adjust the importance of source nodes based on the uncertainty and unstability (which we refer to as difficulty metrics) of their K nearest target nodes. Meanwhile, to mitigate the influence of outliers or unrepresentative samples in the target domain, we employ ego graph-based distance to ensure structural similarity between source nodes and target nodes. In this section, we utilize a first-order logic (FOL) framework [17] to inject structured domain knowledge to balance the difficulty and distance metrics. The core idea is to emphasize representative source nodes whose distribution closely aligns with the target nodes, particularly the challenging target ones.

Here, we use the uppercase $W$ to denote our focused weight term, with the subscript indicating the specific metric being represented. The lowercase $w$ is used to denote the corresponding value. To achieve balance between difficulty and distance metrics, we formulate this structure knowledge as the following FOL rules:

$$(W_{dis} \leq \alpha_1) \wedge (W_{\text{uns}} \geq \alpha_2 \vee W_{\text{unc}} \geq \alpha_3) \qquad (14)$$

Here $\alpha_1, \alpha_2, \alpha_3$ denotes are the thresholds. However, the above formulation is not differentiable. To address it, we define the following indicator function $\psi_g$ and $\psi_l$ for $w_i \geq \alpha$ and $w_i \leq \alpha$ respectively as follows:

$$\psi_g(x_i, \alpha) = \frac{1}{e^{-(w_i - \alpha)}} \tag{15}$$

$$\psi_l(x_i, \alpha) = \frac{1}{e^{(w_i - \alpha)}} \tag{16}$$

Further, motivated by Łukasiewicz Tnorm and T-conorm [16], we relax the logic rules to define a soft version of conjections and disjunctions. Specifically, we follow [17] define the mapping function $\Phi$ to map the outputs of FOL into real values:

- $\Phi(W_i) = w_i$
- $\Phi\left(\bigvee_i W_i\right) = \min\left(1, \sum_i w_i\right)$
- $\Phi\left(\bigwedge_i W_i\right) = \max\left(0, \sum_i w_i - |W| + 1\right)$

The first principle maps a variable in FOL to a real value in the range [0, 1]. The last two principles map the conjunctions and disjunctions to real values in the range [0, 1].

Given a source node $v_i^s$, and one of its nearest target node $v_j^t \in \mathcal{A}_i$. We can calculate the source weight $w_{ij}^s$ via the soft logic of rules in Eq. (14) as follows:

$$w_{ij}^s = \Phi\left(\psi_l(d_{ij}, \alpha_1) \wedge \Phi\left(\psi_g(\hat{w}_{j,uns}^t, \alpha_2) \vee (\psi_g(\hat{w}_{j,unc}^t, \alpha_3))\right)\right) \tag{17}$$

Furthermore, we combine the source weight across nearest target set $\mathcal{A}_i$ to obtain the final value:

$$w_i^s = \frac{1}{K} \sum_{j \in \mathcal{A}_i} w_{ij}^s \tag{18}$$

### 4.3 Model Training

As summarized in the general form of the objective function in Section 3.2, our method can be integrated with existing graph domain adaptation methods. The key difference is that our approach replace $\mathcal{L}_{cls}^s$ with $\mathcal{L}_{cls}^{\hat{s}}$, while keeping other terms unchanged. Specifically, $\mathcal{L}_{cls}^{\hat{s}}$ is defined as follows:

$$\mathcal{L}_{cls}^{\hat{s}} = - \sum_{v_i^s \in V^{\hat{s}}} w_i^s \cdot y_i^s \log \hat{y}_i^s \tag{19}$$

Here, $y_i^s \in Y^s$ is the true label of source node $v_i^s$ and $\hat{y}_i^s$ denotes the model prediction. The acquisition of set $V^{\hat{s}}$ is described in Section 4.1 and the calculation of the source node weights $w^s$ is detailed in Section 4.2. The pseudocode can be found in Appendix A.

## 5 Experiments

### 5.1 Experiment Settings

**Datasets.** Following [22, 33], we conduct experiments on three commonly used real-world datasets provided by ArnetMiner [27]: ACMv9, Citationv1, and DBLPv7, which are from different sources and consequently have varied data distributions. We include six transfer scenarios: C⇒A, D⇒A, A⇒C, D⇒C, A⇒D, and C⇒D, where ACMv9, Citationv1, and DBLPv7 are represented as A, C,

and D for simplicity. We place more details about datasets in Appendix B.

**Metrics.** Following [5], we choose Micro-F1 score and Macro-F1 score to evaluate classification performance.

**Baselines.** We compare our approach against the following baselines, which can be categorized into three categories:

- **Vanilla GNN methods**: GCN [14], GSAGE [12] and GIN [36]. These classical GNN models are used for single graph representation learning. To align them with our settings, following the method in [6], we adapt them by incorporating an additional cross-entropy loss term calculated from the limited labeled target nodes.
- **Unsupervised graph domain adaptation methods**: CDNE [23], ACDNE [22], UDAGCN [33] and MFRReg [39]. These methods are designed for unsupervised graph domain adaptation methods. To align with our settings, we incorporate the classification loss on labeled target nodes into their origin loss functions.
- **Few-shot graph domain adaptation methods**: AdaGCN [5] and SemiGCL [35]. These methods can directly deal with FSGDA task.

**Implementation details.** GraphInflu can be integrated with current graph domain adatation methods. We choose SemiGCL [35] as our backbone which involves cross-entropy loss, contrastive loss and entropy loss. These losses are computed based on identified source nodes $V^{\hat{s}}$, with the classification loss being reweighted accordingly. The target graph is designed to include five labeled nodes for each class. For supportive node selector, we set the training intervals $T = 5$ and a scaling factor $\beta = 1.2$. We set the default selection ratio of source nodes as 0.3. In the node reweighting phase, we set the number of samples to $K = 5$, and thresholds to $\alpha_1 = 0.4$, $\alpha_2 = 0.6$, and $\alpha_3 = 0.6$. The Adam optimizer is used with a default learning rate of $3 \times 10^{-3}$. For a fair comparision, our method and all baselines set the hidden size of feature extractor with 256.

### 5.2 Performance Comparision

We conduct the experiments using 30% of the source nodes and all labeled target nodes for training across all methods. The results for FSGDA on six domain adaptation scenarios, are detailed in Table 1. From this table, we can conclude the following discoveries:

- Compared to the three baseline categories, our proposed method consistently outperforms them, achieving an average improvement of **2.01%** in micro-F1 and a **2.11%** improvement in macro-F1 over the best baseline. This result indicates that, given the same selection budget for source nodes, our method effectively identifies the most influential nodes and prioritizes these core nodes, resulting in enhanced performance.
- Unsupervised graph domain adaptation methods ($LT$ variants) and few-shot graph domain adaptation approaches typically outperform standard GNN methods, emphasizing the critical role of adaptation strategies. However, existing few-shot graph domain adaptation methods underestimate the value of labeled

**Table 1: The model performance is evaluated across six domain adaptation scenarios under a 5-shot learning setting. All methods are trained with 30% of the source data and all labeled target data. The best results are highlighted and the second-best results are underlined. A: ACMv9; C:Citationv1; D: DBLPv7. A⇒C represents that A is the source graph and C is the target graph. The same applies to other scenarios. The methods with the subscript $LT$ represent variants of the baselines that incorporate additional classification loss on labeled target nodes.**

| Methods | C⇒A | | D⇒A | | A⇒C | | D⇒C | | A⇒D | | C⇒D | |
|---|---|---|---|---|---|---|---|---|---|---|---|---|
| | Micro | Macro | Micro | Macro | Micro | Macro | Micro | Macro | Micro | Macro | Micro | Macro |
| $GCN_{LT}$ | 73.98 | 73.37 | 69.25 | 68.90 | 76.66 | 74.52 | 75.16 | 72.1 | 70.84 | 66.92 | 74.11 | 72.24 |
| $GSAGE_{LT}$ | 69.33 | 67.78 | 64.69 | 62.79 | 72.27 | 70.79 | 74.07 | 71.90 | 69.12 | 66.41 | 71.50 | 69.01 |
| $GIN_{LT}$ | 72.87 | 72.50 | 68.74 | 67.80 | 75.53 | 73.98 | 72.39 | 68.68 | 70.18 | 67.56 | 73.05 | 70.40 |
| $UDAGCN_{LT}$ | 76.98 | 76.64 | 74.02 | 73.99 | 81.30 | 78.72 | 80.40 | 78.54 | 74.73 | 70.72 | 76.67 | 73.66 |
| $MFRReg_{LT}$ | 73.48 | 74.98 | 73.57 | 75.69 | 81.53 | 80.23 | 81.82 | 80.56 | 74.80 | 72.80 | 77.48 | 74.70 |
| $ACDNE_{LT}$ | 73.15 | 74.42 | 69.09 | 67.81 | 81.20 | 79.82 | 79.13 | 75.80 | 74.55 | 72.74 | 76.25 | 72.85 |
| AdaGCN | 73.54 | 73.32 | 70.23 | 69.50 | 77.10 | 75.11 | 75.08 | 72.90 | 72.07 | 68.71 | 73.80 | 70.94 |
| SemiGCL | 77.73 | 77.48 | 75.98 | 75.27 | 83.25 | 81.63 | 82.79 | 81.25 | 74.35 | 72.31 | 77.35 | 75.61 |
| GraphInflu | **79.86** | **79.91** | **79.87** | **79.48** | **84.52** | **82.83** | **84.14** | **82.43** | **76.89** | **75.44** | **78.79** | **77.01** |
| Improv.(%) | +2.13 | +2.43 | +3.89 | +3.79 | +1.27 | +1.20 | +1.35 | +1.18 | +2.09 | +2.64 | +1.31 | +1.40 |

**Table 2: The model performance comparison across six domain adaptation scenarios under a 5-shot learning setting. All baselines employ the entire labeled source data and all labeled target data. Other captions are consistent with the Table 1.**

| Methods | C⇒A | | D⇒A | | A⇒C | | D⇒C | | A⇒D | | C⇒D | |
|---|---|---|---|---|---|---|---|---|---|---|---|---|
| | Micro | Macro | Micro | Macro | Micro | Macro | Micro | Macro | Micro | Macro | Micro | Macro |
| $GCN_{LT}$ | 75.31 | 75.31 | 71.74 | 71.46 | 77.72 | 75.93 | 76.72 | 73.75 | 70.72 | 68.42 | 75.08 | 72.90 |
| $UDAGCN_{LT}$ | 78.52 | 78.54 | 75.53 | 74.26 | 81.08 | 78.82 | 81.18 | 79.58 | 75.42 | 72.26 | 77.36 | 74.71 |
| SemiGCL | 78.54 | 78.47 | 77.22 | 77.05 | 83.49 | 81.50 | 83.29 | 81.41 | 75.65 | 73.39 | 77.73 | 75.88 |
| GraphInflu | 79.86 | 79.91 | 79.87 | 79.48 | 84.52 | 82.83 | 84.14 | 82.43 | 76.89 | 75.44 | 78.79 | 77.01 |

target nodes, leading to performance that is on par with traditional graph domain adaptation methods ($LT$ variants). In contrast, our approach effectively utilize labeled target nodes to identify the core supportive data from source domain.

**Comparison with baselines using all source nodes.** To further illustrate the effectiveness of our approach, we compare our model using 30% of the source nodes against the baselines that utilize all labeled source nodes. Additionally, all methods utilize the labeled target nodes. We select the three representative methods from three types of baselines. As illustrated in Table 2, our model's performance still exceeds that of the baseline models. This result demonstrates that not all source nodes contribute equally to generalization in the target domain. Our method is able to identify high-value nodes, thereby facilitating the domain adaptation process. We include more baselines results in Appendix C.

**Performance under 1-shot labeled nodes for each class.** We further explore the performance of our approach under extreme conditions, specifically with 1-shot labeled nodes for each class. This experiment aims to determine whether our method can effectively identify valuable information to help pinpoint the most

valuable source nodes. We present our findings in Table 3. Despite the extremely limited information, our approach demonstrates a significant performance improvement compared with other baselines, showcasing its robustness. Meanwhile, we observe a degradation in performance compared to Table 1, likely due to the fact that a single sample may originate from outliers in the target domain. In practice, we find that using 5-shot labeled nodes achieves satisfactory performance, striking a balance between performance and label costs.

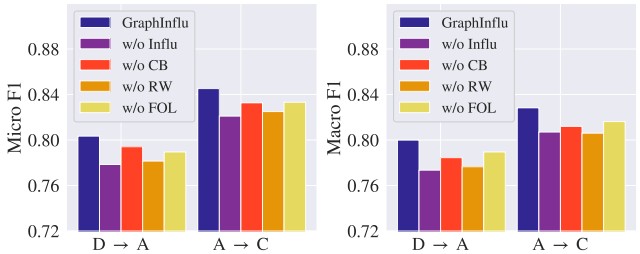

**Figure 3: Ablation studies on D ⇒ A and A ⇒ C.**

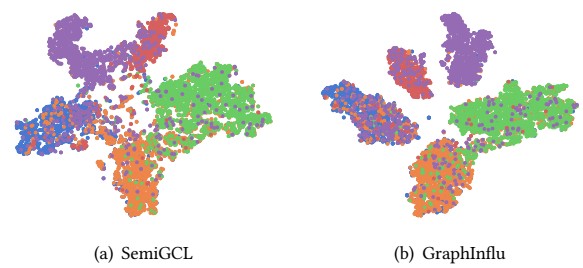

(a) SemiGCL                    (b) GraphInflu

**Figure 4: Visualization of node representations learned by SemiGCL and GraphInflu in the scenario D ⟹ A.**

**Table 3: Performance on 1-shot labeled nodes for each class in the scenarios D ⟹ A and A ⟹ C.**

| Methods | | $GCN_{LT}$ | $UDAGCN_{LT}$ | SemiGCL | GraphInflu |
|---|---|---|---|---|---|
| D ⟹ A | Micro | 69.97 | 71.22 | 74.42 | 76.18 |
| | Macro | 67.99 | 69.02 | 73.15 | 74.60 |
| A ⟹ C | Micro | 76.83 | 80.45 | 82.42 | 83.45 |
| | Macro | 74.17 | 77.02 | 80.35 | 81.60 |

## 5.3 Ablation Study

To validate the effectiveness of components, ablation studies are conducted on:

- **w/o Influ**: This variant disregards the calculation of the contribution score and randomly selects the source nodes.
- **w/o CB**: The class-balanced selection strategy is omitted, and source nodes are selected solely based on score function.
- **w/o RW**: All source nodes are treated equally, with their weights in the classification loss set to one.
- **w/o FOL**: The FOL framework is removed, and source node weights are assigned only based on the product of the difficulty and distance metrics.

The results are depicted in Figure 3. The superior performance of GraphInflu over its variant **w/o Influ** demonstrates the importance of contribution score function. This is attributed to the fact that the few labeled target nodes provide informative guidance in the form of gradient features, which facilitate the selection of source nodes. The performance drop of the variant **w/o CB** indicates that unbalanced source nodes can bias the model towards a suboptimal optimization direction. Furthermore, the variant **w/o RW** underperforms GraphInflu, implying that source nodes do not contribute equally to the domain adaptation on the target graph. The refined reweighting design encourages the model to focus on representative and challenging nodes, which can contribute to performance improvement once addressed. In addition, the observed decrease in performance of the variant **w/o FOL** highlights the effectiveness of our soft logic rules in combining various conditions, including representativeness metric and difficulty metric.

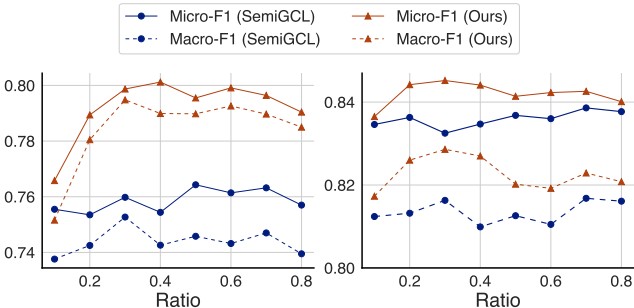

**Figure 5: The effect of source ratio in the scenarios D ⟹ A (left) and A ⟹ C (right).**

## 5.4 More Analysis

**Visualization of representative space.** We compare the representation distributions generated by our model and SemiGCL. Specifically, we employ t-SNE [29] to project the learned node representations in the target domain into a two-dimensional space, using different colors to denote different classes. As shown in Figure 4, the representations learned by our approach form more concentrated clusters compared to SemiGCL. This indicates that GraphInflu can effectively identify the influential source nodes that significantly contribute to the adaptation process in the target domain.

**Effect of selection ratio.** In this experiment, we further investigate the impact of selecting different proportions of source nodes on the target graph's performance. We select SemiGCL (the optimal baseline) for comparison with our method. As illustrated in Figure 5, our approach consistently demonstrates strong performance across varying source data ratios. Notably, our method quickly achieves optimal performance, followed by a gradual decline as the source node ratio increases. This quick increase can be attributed to the effective identification of high-value nodes from the source domain. However, as the ratio of source nodes continues to rise, the model's performance is inevitably impacted by bias caused by adverse nodes. This demonstrates that the performance improvement primarily originates from the supportive nodes, while the improper inclusion of more adverse nodes leads to model degradation. In contrast, SemiGCL's performance remains relatively stable across different source node ratios, suggesting its lower effectiveness in identifying and prioritizing the most valuable nodes.

## 6 Conclusion

This paper investigates few-shot graph domain adaptation for node classification. We argue that the indiscriminate use of all available source nodes limits the GNN performance on the target network. This limitation primarily arises from source-domain bias misaligning with the target domain and the presence of low-value nodes that hinder model training. To address this issue, we propose a novel method, GraphInflu, which introduces two modules: the Supportive Node Selector and Soft Logic-Inspired Node Reweighting. Experiments demonstrate the superior performance of GraphInflu by surpassing state-of-the-arts. Extensive experiments including the ablation study prove the reasonable design of each module.

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

# A  Algorithms Details

Algorithm 1 and Algorithm 2 show the pseudocodes of *Supportive Node Selector* and *Soft Logic-Inspired Node Reweighting*, respectively.

---

**Algorithm 1:** Supportive Node Selector

---

**Input:** source graph $G^s$, target graph $G^t$, training intervals $T$, epochs $E$, sample number $Q$, number of classes $C$, scaling factor $\beta$

**Output:** supportive source node set $V^{\hat{s}}$

Initialize a single-domain node classifier $g_\theta$

Initialize score matrix $\Gamma = \mathbf{0}_{n^{s,l} \times n^{t,l}}$

Initialize supportive source node set $V^{\hat{s}} = \emptyset$

// Calculate contribution score for source nodes

**for** $e = 1$ **to** $E$ **do**

    Train GNN model $g$ using source graph $G^s$

    **if** $e\%T == 0$ **then**

        Calculate current update step $k = \frac{e}{T}$

        Calculate source node gradient store $\nabla_{\theta_k} \mathcal{L}(v^s, \theta_k)$

        Calculate target node gradient store $\nabla_{\theta_k} \mathcal{L}(v^t, \theta_k)$

        Obtain score matrix $\Gamma^k$ using Eq. (3)

        Update score matrix $\Gamma = \Gamma + \eta_k \cdot \Gamma^k$

// Class-balanced selection strategy

Calculate class-wise contribution score $\hat{\Gamma}$ using Eq. (5)

Calculate the number of samples for each class $n = \frac{(1+\beta)Q}{C}$

**for** $c = 0$ **to** $C - 1$ **do**

    Obtain class-wise node set $V_c^s = \text{Top-n}\left(\hat{\Gamma}[:, c]\right)$

    $V^{\hat{s}} = V^{\hat{s}} \cup V_c^s$

---

---

**Algorithm 2:** Soft Logic-Inspired Node Reweighting

---

**Input:** Selected source graph $G^{\hat{s}}$, target graph $G^t$, sample number $K$, GNN model $g$, thresholds $\alpha_1, \alpha_2, \alpha_3$, steps $E$

**Output:** source node weight $w^s$

Obtain representation $z^s, z^t$ and prediction $p^t = h(z^s)$

Choose $K$ nearest samples for source nodes $\mathcal{A}$ with

    $\mathcal{A}_i = \text{Top-K}_j \left( \max \left( \cos(z_i^s, z_j^t) \right) \right)$

Calculate the ego graph based distance $d$ using Eq. (8)

// Unstability estimate

Initializ a random perturbation maxtrix $\Delta$

**for** $e = 1$ **to** $E$ **do**

    Calculate adversarial perturbation $A'$ using Eq. (9)

    Calculate eigvalue matrix $\Lambda = \text{Eig}(\text{Laplacian}(A'))$

    Calculate spectral distance $D_{\text{spec}}$ using Eq. (10)

    Update $\Delta$ with graident descent

Calculate unstability $\hat{w}_{uns}^t$ using Eq. (12)

Calculate uncertainty $\hat{w}_{unc}^t$ using Eq. (13)

Calculate logic forms: $\phi_l(d, \alpha_1), \phi_g(\hat{w}_{unc}^t, \alpha_2), \phi_g(\hat{w}_{uns}^t, \alpha_3)$

Calculate the source weight $w^s$ using Eq. (17)

---

**Table 4: Statistics of the three datasets. '#' indicates the number of instances.**

| Dataset | #Nodes | #Edges | #Attributes | #Union Attributes | #Labels |
|---|---|---|---|---|---|
| ACMv9 | 9,360 | 15,602 | 5571 | 6,775 | 5 |
| Citationv1 | 8,935 | 15,113 | 5379 | 6,755 | 5 |
| DBLPv7 | 5,484 | 8,130 | 4412 | 6,755 | 5 |

# B  Dataset Details

We summarize the dataset statistics in Table 4. These three datasets are citation networks from different sources: DBLP (from 2004 to 2008), ACM (after 2010), and Microsoft Academic Graph (before 2008). Consequently, they have varied data distributions. We represent each citation network as an undirected graph where each node denotes a paper, and an edge corresponds to a citation relationship between two papers. Each paper belongs to one of the following five categories based on its research topics: *Artificial Intelligence*, *Computer Vision*, *Database*, *Information Security*, and *Networking*.

# C  More Experiments

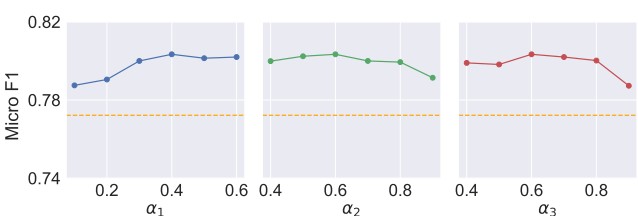

**Figure 6: The model performance with thresholds $\alpha_1, \alpha_2, \alpha_3$ on D ⇒ A. The dashed line indicates the performance of the best baseline model.**

**Effect of thresholds.** Figure 6 shows the effects of the three thresholds $\alpha_1$, $\alpha_2$ and $\alpha_3$. According to Eq. 14, $\alpha_1$ represents the constraint we impose to ensure that the source node is close to the target domain. Meanwhile, $\alpha_2$ and $\alpha_3$ ensure that the corresponding target nodes present challenges based on unstability and uncertainty metrics. We observe that the model's performance increases and remains stable as $\alpha_1$ increases. In contrast, for $\alpha_2$ and $\alpha_3$, there is an opposite trend: the model's performance remains relatively stable at low values but degrades as these values become excessively large. From Eq. 14, we can infer that smaller values of $\alpha_1$ combined with larger values of $\alpha_2$ and $\alpha_3$ lead to stricter logic rules. This causes the model to treat all source nodes equally, failing to provide a nuanced distinction in the contributions of individual source nodes.

**Comparison with baselines using all labeled nodes.** We present additional baselines using all labeled source nodes in Table 5.

Received 20 February 2007; revised 12 March 2009; accepted 5 June 2009

**Table 5: The model performance comparison across six domain adaptation scenarios under 5-shot setting. All baselines employ the entire labeled source data and all labeled target data. The best results are highlighted and the second-best results are underlined. A: ACMv9; C:Citationv1; D: DBLPv7. A⇒C represents that A is the source graph and C is the target graph. The same applies to other scenarios. The methods with the subscript $LT$ represent variants of the baselines that incorporate additional classification loss on labeled target nodes.**

| Methods | C⇒A | | D⇒A | | A⇒C | | D⇒C | | A⇒D | | C⇒D | |
|---|---|---|---|---|---|---|---|---|---|---|---|---|
| | Micro | Macro | Micro | Macro | Micro | Macro | Micro | Macro | Micro | Macro | Micro | Macro |
| $GCN_{LT}$ | 75.31 | 75.31 | 71.74 | 71.46 | 77.72 | 75.93 | 76.72 | 73.75 | 70.72 | 68.42 | 75.08 | 72.90 |
| $GSAGE_{LT}$ | 70.58 | 69.65 | 67.50 | 66.64 | 73.29 | 71.34 | 71.69 | 69.14 | 67.63 | 65.21 | 71.48 | 69.06 |
| $GIN_{LT}$ | 73.09 | 72.28 | 70.75 | 69.29 | 77.21 | 75.19 | 74.72 | 72.81 | 71.21 | 65.30 | 73.69 | 70.85 |
| $UDAGCN_{LT}$ | 78.52 | 78.54 | 75.53 | 74.26 | 81.08 | 78.82 | 81.18 | 79.58 | 75.42 | 72.26 | 77.36 | 74.71 |
| $MFRReg_{LT}$ | 75.29 | 76.85 | 75.28 | 76.71 | 82.44 | 81.10 | 82.12 | 80.91 | 75.69 | 73.29 | 77.75 | 75.61 |
| $ACDNE_{LT}$ | 74.57 | 75.91 | 72.58 | 73.84 | 81.23 | 79.81 | 79.77 | 78.21 | 74.24 | 71.17 | 74.26 | 70.39 |
| AdaGCN | 74.87 | 74.82 | 72.51 | 71.86 | 77.53 | 75.33 | 77.29 | 75.27 | 71.50 | 68.20 | 75.05 | 71.40 |
| SemiGCL | 78.54 | 78.47 | 77.22 | 77.05 | 83.49 | 81.50 | 83.29 | 81.41 | 75.65 | 73.39 | 77.73 | 75.88 |
| GraphInflu | **79.86** | **79.91** | **79.87** | **79.48** | **84.52** | **82.83** | **84.14** | **82.43** | **76.89** | **75.44** | **78.79** | **77.01** |
| Improv.(%) | +1.32 | +1.37 | +2.65 | +2.43 | +1.03 | +1.33 | +1.25 | +1.02 | +1.20 | +2.05 | +1.04 | +1.13 |

