# OpenReview forum: "Grasp the Key Takeaways from Source Domain for Few Shot Graph Domain Adaptation"
_ACM.org/TheWebConf/2025/Conference — WWW 2025 Poster_

### Official Review · Reviewer_79YH · 2024-11-16

**Novelty:** 3
**Technical Quality:** 4

**Review:**

This paper proposes a novel method, GraphInflu, to address the problem of Few-Shot Graph Domain Adaptation (FSGDA). Its core objective is to optimize graph neural network training in target domains with limited labeled data by selecting and re-weighting key nodes from the source domain.

# Cons
1. Graph domain adaptation is an urgent and challenging issue that remains underexplored.
2. The authors have open-sourced their code, which enhances reproducibility.

# Pros
1. Many recent and representative works in this field have not been analyzed or discussed. The comparative methods used in this paper are relatively outdated, mostly limited to works from 2022. Furthermore, the related work section does not provide a clear overview of the latest advancements, making it difficult to identify the shortcomings of current approaches. This is the most significant drawback of the paper.
2. The proposed method introduces numerous approximations, which result in higher complexity and reduced scalability.
3. What are the differences between the proposed node re-weighting strategy and StucRW (ICML23)?
4. The datasets used in the experiments are limited to only three citation graphs. Commonly used datasets like Cora, Cornell, and Arxiv are notably absent.

[1] Semi-supervised Domain Adaptation in Graph Transfer Learning,IJCAI23

[2] Rethinking Propagation for Unsupervised Graph Domain Adaptation, AAAI24

[3] Multi-source Unsupervised Domain Adaptation on Graphs with Transferability Modeling,KDD24

[4] Structural Re-weighting Improves Graph Domain Adaptation,ICML23

**Questions:**

See above questions

**Reviewer Confidence:**

3: The reviewer is confident but not certain that the evaluation is correct

**Scope:**

3: The work is somewhat relevant to the Web and to the track, and is of narrow interest to a sub-community

---

### Official Review · Reviewer_inq8 · 2024-12-01

**Novelty:** 5
**Technical Quality:** 5

**Review:**

This paper introduces GraphInflu for few-shot graph domain adaptation, aiming to identify high-value nodes while filter out adverse source nodes and further enhance the importance of these high-value nodes. The experiments demonstrate that the proposed algorithm outperforms existing baselines in scenarios of few-shot graph domain adaptation.

**Pros**:
1. The authors observed the counterbalance phenomenon between “beneficial” and “harmful” samples.

2. This paper proposed GraphInflu that identifies and reweighs high-value nodes, thus improving the performance.

3. Extensive experiments are conducted to demonstrate the effectiveness of proposed method.

**Cons**:
1. The experimental evaluation is limited to citation networks. For a more generalizable approach, a broader range of datasets should be explored.

**Questions:**

1. The authors state that using 5-shot labeled nodes achieves a balance between performance and labeling cost. They should provide experimental evidence to substantiate this claim.

**Reviewer Confidence:**

3: The reviewer is confident but not certain that the evaluation is correct

**Scope:**

4: The work is relevant to the Web and to the track, and is of broad interest to the community

---

### Official Review · Reviewer_xUGq · 2024-12-02

**Novelty:** 5
**Technical Quality:** 5

**Review:**

The authors introduce GraphInflu, a novel few-shot method designed to tackle the Graph Domain Adaptation problem, particularly in contexts with scarce target label availability.

The framework leverages on Graph Neural Networks (GNNs) through two key modules:

1) Supportive Node Selector: a GNN model identifies influential source nodes by analyzing gradients from both source and target graphs, computing a contribution score. A class-balanced sampling strategy ensures fair selection across classes.

2) Soft Logic-Inspired Node Reweighting: a second GNN model refines the selected nodes by combining structural similarity (ego-graph analysis) and difficulty metrics (instability and uncertainty). These metrics are integrated into a First-Order Logic (FOL) framework for structured and effective reweighting.
The authors conducted extensive experiments on three benchmark datasets across six adaptation scenarios. The results consistently demonstrate that GraphInflu outperforms state-of-the-art baselines, showcasing its effectiveness and robustness. Overall, the authors provide a comprehensive description of the methodology. The methodology and results are valuable, but some revisions could improve the clarity and readability of the paper.

Strong Points
- Innovative approach: the authors present a novel framework that effectively addresses key challenges in Few-Shot Graph Domain Adaptation.
- Comprehensive evaluation: the method is rigorously tested on three benchmark datasets across six adaptation scenarios, consistently achieving superior results compared to state-of-the-art baselines.
- Reproducibility: it is possible to reproduce the results since the source code is provided by the authors.

Weak Points

- the authors analyze the impact of varying source node ratios on model performance (Figure 5), observing a performance decline at higher ratios due to the inclusion of adverse nodes. While this analysis is insightful, it would be valuable to include results for the scenario where 100% of the source nodes are used. This would not only clarify whether the proposed method maintains its effectiveness in the presence of a larger proportion of adverse nodes but also provide a more direct comparison with baseline methods, which use all available source nodes.

- some grammatical issues reduce the paper's readability. Examples include:

1) "The representativeness metric is evaluated based on the distance between ego graphs centered specific nodes from both domains" (sect. 4) is unclear, maybe "The representativeness metric is evaluated based on the distance between ego graphs centered on specific nodes from both domains."
2) "gradient stores" (sect. 4) is also unclear, maybe "stored gradients" or "gradient storage"?
3) "the degrades model performance" (sect. 4.1) should be "this degrades model performance" or "the degraded model performance"
4) "we follow [17] define the mapping function" (sect. 4.2) should be "we follow [17] to define the mapping function."
5) few shot -> few-shot

Minor:
1) "KL" (Kullback-Leibler divergence) is not defined at first use in Section 4, but it is introduced later in sect. 4.2. A reference might help.
2)"LT variants" in sect. 5.2 is not defined.

**Questions:**

As highlighted in the Weak Points, the analysis excludes results for 100% source nodes. Could the authors clarify how the method performs when using all source nodes, especially in comparison to baselines that utilize the full set?

**Reviewer Confidence:**

3: The reviewer is confident but not certain that the evaluation is correct

**Scope:**

4: The work is relevant to the Web and to the track, and is of broad interest to the community

---

### Official Review · Reviewer_3CwF · 2024-12-03

**Novelty:** 4
**Technical Quality:** 5

**Review:**

Pros:

(1)According to the authors, they uncover the substantial negative impact of adverse source nodes when the annotated target nodes are scarce, and they delve into early exploration from the view of the key source-node retrieval for the Few Shot Graph Domain Adaptation.

(2)They propose a novel method named GraphInflu, which filters out adverse source nodes and subsequently enhances the importance of high-value nodes.

(3)Experiments demonstrate the effectiveness and superiority of the proposed method.

Cons:

(1)There are too many notations in Figure 2, which to some extent affects the readability.

(2)Although evaluation methods have been provided, it remains not clear which nodes are unstable and uncertain target nodes intuitively.

(3)The efficiency analysis and corresponding efficiency experiments of the two proposed modules, i.e., the Supportive Node Selector and Soft Logic-Inspired Node Reweighting are missing.

**Questions:**

(1)Can the authors further explain which target nodes are easy to be unstable and uncertain intuitively? And why do these nodes exist？This will further strengthen the quality of the manuscript.

(2)Can the authors supplement the efficiency experiments of the two proposed modules and the traditional model training, to check weather the proposed modules additionally bring significant time consumption？

**Reviewer Confidence:**

3: The reviewer is confident but not certain that the evaluation is correct

**Scope:**

3: The work is somewhat relevant to the Web and to the track, and is of narrow interest to a sub-community